# Impact and Effectiveness of Group Strategies for Supporting Breastfeeding after Birth: A Systematic Review

**DOI:** 10.3390/ijerph18052550

**Published:** 2021-03-04

**Authors:** Isabel Rodríguez-Gallego, Fatima Leon-Larios, Isabel Corrales-Gutierrez, Juan Diego González-Sanz

**Affiliations:** 1Virgen del Rocío University Hospital, Red Cross Nursing University Centre, University of Seville, 41009 Seville, Spain; isroga@cruzroja.es; 2Nursing Department, Faculty of Nursing, Physiotherapy and Podiatry, University of Seville, 41009 Seville, Spain; 3Surgery Department, Medical School, University of Seville, 41009 Seville, Spain; icorrales@us.es; 4Foetal Medicine Unit, Virgen Macarena University Hospital, 41009 Seville, Spain; 5COIDESO Research Center, Nursing Department, University of Huelva, 21071 Huelva, Spain; juan.gonzalez@denf.uhu.es

**Keywords:** breastfeeding, lactation, self-help group, support group

## Abstract

Despite the multiple benefits of breastfeeding both for the mother and for the infant, during the first months there is a progressive decline in the number of mothers who continue breastfeeding, with most countries reporting lower than recommended figures. The objective of this review is to analyse the most effective group support practices for breastfeeding, as well as the characteristics associated to their success in maintaining breastfeeding. A systematic review has been conducted in the 2015–2020 period, in the following databases: MedLine, Scopus, Web of Science, Cumulative Index to Nursing and Allied Health Literature, Cochrane Library and LILACS. The findings were summarized in narrative and table formats. A total of thirteen articles were included, eight of them being experimental studies and five observational. The findings show high heterogeneity regarding the characteristics of the interventions and their periodicity. The most successful group strategies to support and maintain breastfeeding during postpartum are those that combine peer support with the leadership or counselling of a health professional or IBCLC. However, more studies are necessary, randomized and with interventions of similar characteristics, which allow for better data comparison.

## 1. Introduction

The World Health Organization (WHO) advocates breastfeeding (BF) as the most suitable means for the healthy growth and development of lactating children, as an unparalleled nutrition means capable of providing all the energy and nutrients that the infants need in its first months of life, contributing half or even more of the child’s nutritional needs during the second semester of life, and up to one third during the second year [1].

Despite the multiple benefits gathered in the literature both for the mother and for the infant at the physical, cognitive and psychosocial levels [1,2,3,4,5,6], the global situation reflects that, although BF initiation is most common in almost all countries, there is a progressive decline in the number of mothers who continue breastfeeding during the first months of life [7]. In the 2025 nutrition objectives, the WHO (2014) sets forth increasing at least 50% the Exclusive Breastfeeding (EBF) rate during the first 6 months of life. However, the situation in industrialized countries is worrying, way below the recommendations by this organization [2]. According to the report published in 2016 by the United Nations Children’s Fund (UNICEF) [8], it is estimated that, globally, 36% of the children are on EBF until the 6th month of life. In Europe, the data regarding prevalence at six months of life are 16% EBF and 41% mixed BF; the highest rates were found in Southern Asia, with 60%, and in the East and South of Africa, with 57%.

The main problems described which are related with early weaning during the first two weeks of life are the following: perception of insufficient milk production, problems with latching or inadequate suction by the child, appearance of crevices in the nipple or other breast-related problems [9,10]. Likewise, the same difficulties related to BF are also evidenced in the subsequent months, accompanied by other reasons such as insufficient weight gain in the infant or the mother’s return to work. These reasons emerge both in developed societies and in less favoured environments [10,11]. However, there are noticeable differences in early weaning between the women who received support to the detriment of the first ones [9,10], with some women describing that they would have continued breastfeeding if they had received precise information and support [12]. 

In relation to this support, the interventions described in the literature are numerous and very heterogeneous. It is recommended to promote support through the health care system, providing counselling and specialized help in primary care and hospitalization consultations. On the other hand, it is also recommended to encourage breastfeeding support in the community, with collaboration between health professionals and the community-based support networks through breastfeeding workshops and contact with local support groups [2,13,14]. In the last few years, the support interventions conducted through telephone contacts or the use of social networks have also gained special relevance [14,15,16,17,18,19].

Specifically, in relation to the group interventions, the literature describes that they can be conducted by the health services themselves, holding breastfeeding workshops, or by other non-professional instances, such as breastfeeding support groups (BFSGs) or even by breastfeeding counsellors. Despite presenting different structures and hierarchies, they have the common objective of mutually supporting each other and attaining successful breastfeeding, sharing experiences, and achieving personal and social changes through mutual support based on cooperation, always strongly emphasizing personal interaction and that the members assume their responsibilities [20,21,22]. 

The objective of this systematic review is to analyse the most effective breastfeeding group support practices performed during postpartum, as well as the characteristics associated to their success in maintaining BF for a longer period of time. 

## 2. Materials and Methods

A systematic review which adhered to the 2015 Preferred Reporting Items for Systematic Reviews and Meta-Analyses (PRISMA) guidelines [23] (Appendix A), was conducted from November to December 2020. 

The systematic search in the literature was conducted on the following electronic databases: MedLine via Ovid, Scopus, Web of Science (WoS) via Thomson Reuters, Cumulative Index to Nursing and Allied Health Literature (CINAHL) via EBSCO, Cochrane Library via Wiley and LILACS via EBSCO. In addition, relevant grey literature sources including Google and Google Scholar were also searched.

The Population/Intervention/Comparison/Outcomes (PICO) format was used to prepare the research question:

Population: Women exposed to group interventions intended to promote and support breastfeeding. This includes mothers of preterm or term infants, with singleton or multiple births. We also included population subgroups of women, such as women from low-income or ethnic groups. Women and infants with a specific health problem, e.g., women with HIV/AIDS, diabetes, obese women and infants with cleft palate, were excluded from this review.

Intervention: Support during breastfeeding was defined as any strategy targeted at those mothers with the purpose of encouraging breastfeeding and favouring its sustainment for a longer period of time during postpartum. Specifically, peer support and mother-to-mother support groups, with or without leadership from a healthcare professional, has been included. They are small groups of pregnant women and/or mothers who are breastfeeding—or have breastfed—who meet regularly in order to share appropriate experiences, support and information about breastfeeding.

Comparison: Usual care.

Outcome: The main outcome was the breastfeeding rate in the months following birth. Secondary outcomes: characteristics of the support groups.

### 2.1. Search Strategy

The search strategy designed was conducted with the combination of the Medical Subject Headings (MeSH) and free terms thesaurus, by using the Boolean operators AND and OR. The MeSH terms used were the following: “breast feeding”, “lactation”, “self-help group” and “Group Processes”. 

The following search equation was employed: (breastfeeding OR “breast feeding” OR “milk secretion” OR lactation) AND (“support group” OR meeting OR “self-help group” OR “breastfeeding workshop” OR “Group Processes”) AND (impact OR effect OR influence).

Table 1 shows the different search equations applied in each database, with the filters used and the results obtained in each of them.

### 2.2. Inclusion/Exclusion Criteria

The inclusion criteria were as follows: original studies randomised controlled trials (RCTs) or cluster-RCTs, with or without blinding, non-randomized controlled trials, quasi-experimental, experimental trials with no comparison group and observational studies, studies limited to the 2015–2020 period, in English, Spanish and Portuguese.

The exclusion criteria were as follows: studies that dealt with the description of the phenomenon from the social perspective of the support groups or that reported data on the most explored themes were excluded, as well as studies that dealt with postpartum support only from the institutional support without group participation, qualitative studies and series of cases. We also excluded abstracts for which we could not find the full reports and studies with low quality assessments.

### 2.3. Data Extraction

Two of the authors conducted the search by pairs (F.L.-L. & I.R.-G.). The eligible studies recovered from the six bibliographical databases were imported into the Mendeley Software reference manager (London, UK), and duplicates were removed. The search in the grey literature did not provide relevant results. The selection of studies was conducted based on titles and abstracts to determine their relevance. The full texts of the remaining studies were read to determine their eligibility, and those that met the inclusion criteria were maintained. Once the eligibility process was over, two authors (F.L.-L. & I.R.-G.). assessed the methodological quality and the biases of the potentially useful studies; this allowed improving the screening of the results in order to obtain more complete and relevant information, thus enhancing the quality of the study. The agreement degree between the two researchers in terms of evaluating the eligibility of the study was assessed using Kappa’s statistical test, obtaining a high result regarding agreement (Kappa statistics = 0.81).

Due to the high heterogeneity of characteristics of the participating populations, as well as to the interventions conducted, it was not possible to perform a meta-analysis. A narrative analysis of the results was conducted. Information was extracted on the sample characteristics, intervention locus, type, duration and frequency of the intervention, and tools employed for measuring the results. 

### 2.4. Quality Assessment and Risk of Bias

The assessment of the methodological quality of all the articles included in the systematic review was performed by two of the researchers, following the reporting guidelines proposed for the main types of studies by the Enhancing the QUAlity and Transparency Of health Research (EQUATOR) network [24]. Once the articles had been independently analysed by the researchers in their full text and considering the inclusion and exclusion criteria, they were discussed with the other authors until reaching consensus on the articles to be included.

On the other hand, for the assessment of the articles with an observational design, the statement of the Strengthening the Reporting of Observational studies in Epidemiology (STROBE) initiative [25] was used. This is a checklist consisting in 22 items related to the title, abstract and introduction, as well as to the methods, results and discussion sections of the articles. Eighteen items are common to the three study modalities, and four refer specifically to the cohort, case and control or cross-sectional studies (Appendix A). For the studies with an experimental design, the recommendations of the 25-item checklist of the 2010 CONsolidated Standards of Reporting Trials (CONSORT) statement [26] were followed (Appendix A).

After assessing the methodological quality of all the articles according to their design, the observational studies were classified in relation to a percentage system established by the authors taking as a reference the STROBE scale items. They were identified as “High quality” articles when they met more than 75% of the standard criteria proposed, as “Medium-Low quality” if the percentage was between 50% and 75% of the criteria, and as “Low quality” if the percentage of criteria met was below 50%.

The risk of bias assessment tool was applied to the experimental studies according to Cochrane [27], which classifies each type of risk as low, high, or unclear. The types of risk include the following: generation of random sequences, allocation concealment, blinding of the participants and of the personnel, blinding of the results evaluation, data from incomplete results, selective reporting, and other possible bias sources. It is considered that the studies without a high risk of bias in any category are of high quality (1++) and that those with one high risk or two unclear risks are of medium quality (1+). The rest were considered as of low quality (1–). Bias evaluation was performed with the Review Manager software, version 5.3 (Copenhagen (Denmark): The Nordic Cochrane Centre. The Cochrane Collaboration).

## 3. Results

### 3.1. Literature Search Results: Study Selection

The search conducted in the six databases described generated a total of 1364 results, of which 242 were duplicate citations that were removed using the Mendeley^®^ bibliographical reference manager. Subsequently, the titles and abstracts of 1122 articles were examined, considering eligibility regarding the inclusion criteria defined according to the 32-citation PICO framework. A total of 1083 articles were excluded after realising that neither the title nor the abstract was related to the study objectives or because they did not meet inclusion criteria. After 39 full-text readings, a total of 26 articles were rejected: 14 articles that dealt with formal support without group participation, two articles on prenatal support groups, another narrating the description of the support groups phenomenon from a social perspective, two studies that dealt with the themes emerging in the support groups and, finally, after full-text reading, the results of seven articles were not related to the objectives defined for this research. Finally, a total of 13 articles were included in this systematic review. The PRISMA flow diagram is presented in Figure 1.

### 3.2. Risk of Bias

Figure 2 and Figure 3 show the risk of bias graph of the total of experimental studies included, as well as the summary of the bias risk of each article by items, respectively. Regarding selection bias, four studies provided adequate information on the generation of random sequences [28,29,30,31]. With regard to performance bias, only one study provided information on the blinding of the participants and of the personnel [31]. In relation to detection bias, only two studies provided unclear information [30,32], and the others omitted that information [28,29,31,33,34,35]. A total of six articles [28,29,30,31,33,35] provided unclear information on allocation concealment. In relation to withdrawal bias, the risk was considered uncertain or unclear in four of the studies [28,31,33,35], as they did not detail how the losses of participants in follow-up were treated. In relation to the reporting bias risk, the risk of bias due to incomplete result data was low for all the studies, as well as for their selective reporting. With regard to other risks or limitations, moderate risk was considered in four studies [32,33,34,35]: one for presenting a different proportion of comparative groups [33], another for presenting results based on a segmented regression analysis of a time series [32], the third for establishing a comparison between two groups with different samples [34] and, finally, one for collecting data in the same time interval, both by an observer in the case of the intervention group and self-reported by mail in the case of the control group, which could exert some influence on the results [35].

As it can be seen in Figure 2, reviewed authors’ judgements about each risk of bias item presented as percentages across all included studies, the higher percentage of bias risk is determined by the generation of random sequences and by the blinding of the participants and of the evaluators, due to the very nature of the study interventions. 

### 3.3. Characteristics of the Sample

The methodological design of the articles included in this review is of the observational type in five of them (38.46%), and experimental in eight (61.53%), of which two are randomized clinical trials (15.38%) and six are quasi-experimental studies (46.15%).

A total of 23.07% of the articles were published in 2016, and we found the same percentage of publications in 2019 (*n* = 3). 15.38% of the articles (*n* = 2) were published in 2018 and 2020 each and, finally, one article was published in 2017, and the same number in 2015, the time limit for inclusion in the review. 

The sample of the selected studies included mothers and/or lactating children from different European countries (Finland, United Kingdom), as well as from the United States, India, Taiwan, Kenya and Iran; reason why there is an ample sample with diverse ethnic and socioeconomic characteristics. Likewise, the sample of lactating children includes both full term and preterm infants.

Table 2 shows the main characteristics of the articles included: author, year and country, study design, objective, study characteristics, measuring instrument, main results, impact of breastfeeding (percentage or duration), and quality assessment outcome. 

### 3.4. Participants

A total of 34,306 women are the total sample of this systematic review. Eight of the studies described analysed BF support in healthy full term infants [30,31,32,33,34,35,37,38] but another two analysed such impact in preterm infants [29,36], while most of those that explored the impact of the community groups [28,40] and online [39] support groups did not establish any difference between both groups of neonates.

Similarly, there are differences regarding the socioeconomic characteristics of the target population of such support, with five studies specifying and assessing the effectiveness of these interventions in less favoured populations [30,31,33,34,38]. Although all the studies selected in this review include women of legal age to medical services in their samples, only one incorporated specifically young mother aged below 25 years old [32].

### 3.5. Measuring Instruments and Interval

In seven studies, the results were self-reported by means of questionnaires [28,29,34,35,36,38,39], two of them specifically through questionnaires sent by mail [29,36]. In eight studies, the results were obtained by means of interviews based on hetero-applied structured questionnaires [29,30,31,32,33,37,38,40], in one of them through direct observation by the researcher [32], and through telephone contacts in three of them [29,34,40]. Three articles combined both the aforementioned methods [29,34,38].

In all studies included, the main instruments used to conduct data collection were structured questionnaires [28,29,30,31,32,33,34,35,36,37,38,39,40]. Two of them specifically detailed their characteristics: Infant Feeding Survey (150 questions) [36] and a questionnaire with a Likert-type scale [28]. In addition, three of these studies included data collection by means of validated instruments: IFFAS [29,39], BFSE-SF [29,35,39] and NSB [39]. 

In relation to the measuring intervals with respect to the result on BF, there is high heterogeneity, the measuring interval not coinciding in any of the thirteen articles. Five of the 13 articles included in the review [29,32,34,37,40] performed the measurement in the immediate postpartum (first 24 h after birth). Eight of the studies [28,31,32,34,35,36,38,39] performed the measurement during the first month postpartum. Ten articles [28,29,30,31,32,34,35,36,37,40] collected result measurements in the interval between the first two months postpartum and the sixth month postpartum. Eight [28,29,30,33,34,36,37,40] of the thirteen studies performed measurements beyond the six-month period, once complementary feeding was initiated and until the first year of life of the lactating child, with one of them specifically extending the measurements until two years of life [34].

### 3.6. Characteristics of the Interventions

Regarding the characteristics of the support interventions, weekly periodicity stands out in five [28,30,33,34,40] of the thirteen studies analysed, three of the studies [31,35,37] reported monthly periodicity, and the remaining five [29,32,36,38,39] did not report specific information. In relation to the recommended maximum number of participants, three studies detailed this component, establishing it in 8–10 mothers [35], 15 mothers [31], and 12–20 mothers [37]. 

The characteristics of the support described in the studies were very diverse: community support groups [36], training peer support groups [28], support groups for mothers established within the national health strategy and counselled by health professionals with the help of a trained and experienced mother [37], online peer support groups [29,39], peer support service guided by mothers who have breastfed and designed according to the social cognitive theory to foster self-efficacy and empowerment [32], support group guided by a breastfeeding peer counsellor from the area [34], SHGs devoted to promote healthy practices for mothers and infants [30,31,33,38], and peer support groups organized by IBCLCs [28,35,40]. The different interventions are specifically detailed by blocks below:

In relation to the community support groups, only one study was identified analysing this type of support, that by Rayfield et al. [36], where the BF rates of those mothers who had received advice from a professional and attended a community support group were compared to those of the women who had not received any of these interventions. It did not report specific information on the attendance frequency or duration, nor on the characteristics of the groups. 

On the other hand, three studies assessed the effectiveness of the peer support groups [28,35,40] and, in addition, two of them combined this intervention with the leadership of an IBCLC [35,40]. In their study, Jenkins et al. [40] took in the analysis of the “Baby Café” program in BF, comparing two groups of breastfeeding mothers who attended them. The “Baby Café” program included regular weekly periodicity, as well as the presence of an IBCLC, or oversight by an IBCLC when the café is staffed with alternative approved breastfeeding counsellors. Many cafés also offered guest speakers on various themes and Facebook forums where the mothers were socially connected and shared diverse knowledge and experiences on breastfeeding, although the two Cafés used as study unit in these articles did not specify if they included speakers and/or forums. The program being open, the concrete number of meetings was specified, although it was reported in the study that more than 80% of the mothers attended more than five times. Another of the studies was conducted by Lee et al. [35], who proposed a combined intervention between peers and professionals, the groups being organized and guided by an IBCLC. The periodicity of the meetings was established in a first session at one week postpartum and another at 5–6 weeks, lasting one hour per session, combining training support with adaptation to the needs of the group and establishing a maximum number of 8–10 mothers. 

Only the study by Moudi et al. [28] compared the training peer support sessions with the groups guided by health care providers, where the two aforementioned intervention models were compared with respective control groups. These interventions were conducted before delivery at the 36–38 gestational week and, later, with a weekly periodicity during three weeks postpartum. They combined in-person interventions (first and third) with telephone contacts (second and fourth).

Another study [37] of the 13 articles included in the study combined the support groups with counselling by health professionals together with the leadership of an expert mother on BF trained to offer support, this program being called “Mother Support Groups” (MSGs). These groups were characterized by attendance both of pregnant women and of mothers of children below the age of 5. They established a periodicity of two meetings per month with a maximum number of 20 mothers, establishing optimum attendance at 12–20 mothers.

In the same line, by combining the attendance of pregnant and breastfeeding women in the group, the study conducted in 2017 by Schreck et al. [34] is found, detailing a postnatal intervention group guided by a peer counsellor from the area, although hired for such purpose in this case. Attendance was established on a weekly basis, although without following a fixed-content structure, adapting to the mothers’ needs. It is worth noting that the women attending these groups were granted free transportation to the meeting locus, as well as a light lunch; mean attendance to the group was 3,15 times (SD = 9.1, interval: 1–50).

Another of the studies included in this review also took in the idea of peer support guided by women (BPS) with previous experience in BF and training, and hired for such purpose [32]. This is the only study that included a specific sample of young mothers aged below 25 years old. In addition, this support model was based on the social cognitive theory, which had the objective of influencing breastfeeding self-efficacy through empowerment, contributing a model to be followed and positive reinforcement. Support was offered from the 30–34 gestational week until six weeks postpartum, with follow-up being more intense during the first two weeks postpartum, without specifying its frequency. Support was offered in-person or by telephone contacts, according to the women’s needs; in addition, the possibility of conducting a home visit in the first 24–48 hours postpartum was contemplated. The women hired to provide this support were under the periodic supervision of a health professional, whom they could contact whenever they needed to.

Four of the thirteen studies contemplated a specific model of BF support based on SHGs, which have shown high ability to convey positive messages in relation to motherhood and to the health of the infant, as vehicles in the behavioural changes and as a tool that improves access to the health services. As common characteristics of the studies that contemplated support by means of SHGs in these articles [30,31,33,38], their study with less favoured and vulnerable populations stood out, having as objective not only the health intervention but also to promote self-confidence and empowerment in the women. 

In this way, Saggurti et al. [30] conducted a structured intervention that was developed in eight sessions with weekly periodicity, where one of them was specific on BF. Another of the studies that analysed this type of intervention was also carried out in the same year (2018) in India by Ruducha et al. [38]; however, it did not specify the number of group members or the frequency of the meetings. The third of the studies in this line conducted in India was carried out by Hazra et al. [33] in 2019, where it was specified that the self-help groups were guided by a volunteer from the community who acted as a peer-educator, with training on key behaviours on maternal-neonatal health, with a weekly regimen of meetings. The last study that assessed the BF support offered by the SHGs was developed in Kenya in 2020 [31]; specifically, and unlike the previous ones, this intervention was guided by a trained breastfeeding peer-educator, and the sessions were developed with a structured design, a maximum number of 15 mothers, lasting one hour, and weekly periodicity of meetings until six months postpartum. In addition, the authors evaluated this type of self-help group design by comparing it to another that also added the component of conducting a group activity not related to health (making soap for later sale) as a way to improve group adherence and increase home economies. 

Finally, there were two studies that assessed the online support groups [29,39]; Niela-Vilén et al. [29], the second article that specified the inclusion of mothers of preterm infants in the sample, proposed an online intervention through a Facebook closed support group guided by three volunteer mothers, without specific training and who had preterm infants. The group’s participants could also offer help and solve doubts to the other group members; in addition, there was the figure of a midwife as a counsellor to answer specific questions. In this way, the study combined peer support with formal support online. Adherence to the group was not specified, nor frequency or time of participation. The second article is by Robinson et al. [39], where they assessed an online BF support intervention conducted through Facebook as in the previous study although, in this case, exclusively focused on Afro-American women. This paper did assess the frequency and duration of the intervention, establishing visits by the mothers to the online group several times a day during less than six months as the most frequent measurement. 

### 3.7. Impact of the Group Interventions on BF

#### 3.7.1. Breastfeeding Initiation

The BF initiation rates were assessed in six of the studies included in the review [30,32,33,34,35,38]; however, these data were influenced by the prenatal interventions contemplated by these studies in combination to those postpartum, rather than by these in isolation; or rather, they compared prevalence data reported before implementing the intervention by means of national surveys. Five of the six studies [30,32,33,34,38] reported favourable data in this period, which supported the effectiveness of the interventions on the breastfeeding rates.

In the first place, the quasi-experimental study conducted by Scott et al. [33] recorded a significant trend change in the period following the implementation of the support intervention, guided by women with previous experience on BF, in comparison to the data previously reported from national statistics. In this way, the results showed that the prevalence of breastfeeding at birth started to increase by 0.55 percentage points per month (95% CI: 0.10–1.00) (*p* = 0.018) after the introduction of the Breastfeeding Peer Support Service (BPSS), where the previous figures had been static. It is worth noting that this intervention contemplated a first contact at the 30–34 gestational week, along with subsequent follow-up until six months postpartum. Likewise, in the United States, Schreck et al. [34] showed a higher BF initiation rate in the intervention group compared to the control group (*p* < 0.0001), where a prenatal intervention with an IBCLC was combined with a postnatal support group guided by a BF counsellor hired for such purpose.

In the same line, positive results were also observed in the breastfeeding initiation rates in three of the studies that analysed the impact of the SHGs [30,33,38]. Ruducha et al. [38] observed a significantly higher proportion regarding BF initiation in the intervention group when compared to the control group (*p* = 0.001), where positive messages on motherhood, feeding and infant health had been established in the prenatal stage. Likewise, Saggurti et al. [30] observed favourable data for BF initiation, as well as for EBF, in those women who had received support from the SHGs (*p* < 0.001). In the study by Hazra et al. [32], conducted in India and assessing the effect of the self-help groups guided by a volunteer from the community acting as a peer-educator, significant differences were also evidenced in the comparison between both groups (*p* = 0.047). 

On the other hand, in their study with groups organized by IBCLCs, Leet et al. [35] reported similar rates of exclusive breastfeeding in the first week after hospital discharge, with no significant differences between the intervention and control groups (*p* = 0.11), this being the only one of the articles analysed that did not show favourable findings in the breastfeeding assessments after the intervention.

#### 3.7.2. Two Weeks Postpartum

The following data reported were established in the time frame of 10–14 days postpartum, when the group support intervention had already been started and where the first BF abandonment peak is described in the literature. Two studies specifically gathered data in this interval, both finding positive data in BF after implementing the intervention conducted [32,36]. 

The aforementioned article by Rayfield et al. [36] showed specific figures at ten days postpartum, evidencing statistically significant data in favour of the BF rates both in preterm (*p* = 0.006) and in full term (*p* = 0.043) infants after the mothers’ attendance to community support groups, without specifying the frequency or duration of the meetings. On the other hand, after the introduction of the BPSS in the UK, a steady increase of 0.50 percentage points at two weeks was observed (95% CI: 0.15–0.86, *p* = 0.007). By the end of the study period, this translated into an additional 6.6 women per 100 giving birth per month who initiated breastfeeding and an additional 6 women per 100 who were breastfeeding at 2 weeks compared with the pre-intervention period [32].

#### 3.7.3. Between One and Six Months Postpartum

In relation to this postpartum period, six studies contributed very diverse figures in this time frame, with data on BF and EBF prevalence, rate and duration [28,31,32,35,36,40]. Only one of these articles did not show statistically significant differences (*p* = 0.086) (*p* = 0.106) in the two study groups: healthy and preterm infants, respectively [36]. The other five articles contributed favourable data in relation to BF:

In the first place, Moudi et al. [28] did show statistically significant differences regarding the EBF rate in the three comparison groups (peer support, education by health care providers, control) at four and eight weeks respectively (*p* = 0.043, *p* = 0.023), although not reporting statistically significant data in relation to EBF duration at four weeks (*p* = 0.056), this difference once again being significant at eight weeks (*p* = 0.014). On the other hand, the data offered by Lee et al. [35] at six weeks postpartum in relation to EBF also showed statistically significant differences between the women who had attended peer support groups organized by IBCLCs (61%) compared to the control group (39%) (*p* = 0.01).

Scott et al. [32] reported significant data on the percentage increase of BF prevalence at initiation and at two weeks postpartum, in a significant manner as previously mentioned; however, no significant changes in the BF prevalence trend were detected in the six-week interval. 

The study conducted by M´Liria et al. [31] in Kenya at the second month postpartum established a statistically significant difference between the intervention groups and the control group (*p* < 0.01) but not between the two intervention groups (*p* = 0.034). Similar trends were observed at the third and fourth month. Likewise, at the fifth month postpartum, there were significantly higher percentages of lactating children in the intervention groups that were on EBF than in the CG (*p* = 0.01) but there was no significant difference in the EBF rates with respect to what was observed in the intervention groups (*p* = 0.79).

On the other hand, the “Baby Café” model presented by Jenkins et al. [40] exposed the results of the two groups analysed, Melrose Baby Café and San Antonio Baby Café, against the national data declared in the Centres for Disease Control and Prevention (CDCs). The women attending the first of the cafés reported exclusivity rates of 77% at 3 months, while those who attended the San Antonio Baby Café reported an EBF rate of 52%. These figures are 1.64 and 1.10 times higher in comparison to those declared in the CDCs.

#### 3.7.4. Six Months Postpartum

In relation to six months postpartum, eight of the thirteen articles collected BF assessments at this specific time point, all of them showing favourable data in relation to BF [29,30,33,34,37,38,39,40].

One of the studies conducted in Kenya with the MSGs model [28] did not find significant differences in relation to the BF rates between the two groups observed in this time frame (*p* = 0.414), as also was the case of the closed Interned-based peer support intervention conducted through Facebook by Niela-Vilén et al. [29] with mothers of preterm lactating infants, where no improvements were shown in the EBF or overall breastfeeding rates six and twelve months after the online intervention. In the same manner, in the study by Schreck et al. [34], the continuation rate at 6 months or more did not differ between the baseline groups and those following the intervention (*p* = 0.5), but it was in fact higher among the women who also participated in the breastfeeding support group in comparison to those who only participated in the prenatal intervention, even describing that approximately 95% of the women stated that participating in the support group exerted an influence on their decision to continue BF beyond the first six months [34].

The training peer support with the presence of an IBCLC intervention model in the Baby Café meetings [40] reported higher EBF figures in the intervention groups (71% Melrose, 47% San Antonio) against the data presented by the CDCs (25%), the rates being 2.84 and 1.88 times higher in the support groups than in usual care. Although the EBF percentage values reported by the mothers in both groups declined in the 6-12-month interval, this trend was kept, reporting rates 1.88 and 1.55 times higher in comparison to the CDCs.

Regarding the SHGs, contrasting data are shown in the studies that assess this type of support in relation to EBF. On the one hand, Saggurti et al. [30] assessed the effectiveness of the SHGs on the EBF rates at twelve months (*p* < 0.001); in the same line, the study developed by M´Liria et al. [31] observed an increasing EBF trend in the women from the intervention groups compared to the control group during the first month of the intervention. Specifically, at the sixth month postpartum, the women from the intervention groups presented two times more chances of EBF at six months than those from the control group (*p* = 0.004 and *p* = 0.033, respectively). As in the previous months, there was no significant difference (*p* = 0.174) between the two groups, the cumulative EBF rate being shorter in the CG in comparison to the intervention groups (*p* = 0.001). In opposition, the data obtained by Hazra et al. [35] did not show statistically significant differences regarding EBF in the intervention group compared to the control group at six months postpartum (*p* = 0.697).

On the other hand, the data offered by Niela-Vilén et al. [29], who analysed the effect of the closed Internet-based peer support intervention, did not show significant differences regarding the duration of exclusive BMF at six months postpartum (*p* = 0.10). However, the study by Robinson et al. [39], also based on the online group intervention through Facebook, significantly correlated the support provided by means of this tool with the intended duration of breastfeeding (*p* < 0.05), which suggested that the support received by the mothers in the Facebook group could be an important factor related to the duration of breastfeeding.

## 4. Discussion

### 4.1. Summary of the Findings

This systematic review examined the studies published on the support group strategies for breastfeeding during postpartum, as well as their impact on the breastfeeding rates, which allowed having an overview of the interventions conducted to promote BF in this period, as well as assessing their real impact on breastfeeding. 

There is an ample variety of support practices for breastfeeding that can be beneficial, both for full term [30,31,32,33,34,35,37,38] and for preterm [29,36] infants. Likewise, in relation to the socioeconomic characteristics of the populations in which the interventions described were conducted, their effectiveness has been proved both in more developed societies [28,29,32,34,35,36,39,40] and in less favoured ones [30,31,33,37,38], in line with the findings previously cited in the literature [5,41,42].

In this review, the most frequent characteristic and common to all the interventions was the in-person meetings [28,30,31,32,33,34,35,36,37,38,40], although the online sessions are also shown as an effective support alternative for BF [29,39], with no study exploring their combination. Likewise, another important characteristic agreed upon by the authors is that the contents to be developed during the support group sessions must not be structured or static, but adaptive to the group’s needs [28,30,34].

As the most effective strategy, most of the studies have reported the combination of peer support with the leadership or aid of an IBCLC or health care provider, with no significant differences between both types of leadership, the two components being considered as successful in the support groups that must be taken into account [28,32,34,35,40] in the line of recently published studies [43]. However, when the support strategies are performed with vulnerable populations or in rural areas with limited resources, it is indispensable to consider a community leader with similar characteristics to those of the group members and who acts as a peer-educator, so that the health messages are more effective and better reach the target population [28,30,31,33,37,38].

In relation to the online sessions, the most used tool was Facebook, either as groups closed to the study participants [29] or as groups open to the entire community [39], although it is suggested taking into account the popularity of the pages through which the messages are conveyed, as there is evidence that those with fewer followers are related to a lesser impact on health [44]. Another aspect to be considered with respect to the use of the online groups would be their specific target populations, the recommendation being taking into account the preferences of the social networks of the specific groups, adapting the communication channel to them [39]. As in the in-person sessions, the online groups must include some expert in BF [29], considering that the information coming from the Internet is not always true or adapted to the requirements of the target population [44]. This person could act by adapting the information to the group requirements, contrasting the veracity of the information, as a moderator, or by solving any doubts that arise; although their role is yet to be determined, as described in the literature [45,46]. On the other hand, it is also pertinent to value group adherence, participation and use frequency of the online groups, as the women who receive this type of support are not only provided with information thereof but also of the environment and of other online tools [29,39].

In the studies, there is no consensus in relation to whether the support groups by themselves improve the breastfeeding initiation rates after delivery if comparing the intervention groups with the usual care provided to the women. This could be related both to the prenatal promotion and support interventions conducted [32,34,35] and to the fact that, after delivery, women usually present a positive initial predisposition that favours initiating the practice [47], regardless of the strategies that are developed to foster it. It was indeed observed that the BF rates at two weeks postpartum are better in women who receive BF support strategies [48,49], including those of a group nature [30,33,34,36,38], as well as at 6 months postpartum [28,29,30,33,34,36,37,38,39,40]; but no strategy better than other can be identified, as they are all usually effective. However, beyond six months postpartum, the results are not consistent in the studies, with differences observed, which exerts a notorious influence on this period as favourable predictors for BF success in the mother’s own breastfeeding intention, the self-efficacy perceived with respect to breastfeeding, and previous experience [29,39]. 

As a tool to take into account to improve adherence to the support groups, its effectiveness has been proved by the fact of sharing social activities that can be developed in a group manner, in addition to sharing the very experience of breastfeeding [31,34,38,40], being an important incentive to improve the breastfeeding rates in populations with less favoured socioeconomic characteristics, where lower BF rates seem to exist [50]. In more resourceful populations, the fact of receiving support while sharing a meal or a coffee can be a valid resource [34,40]; however, in populations with limited resources or in a situation of marginality, this incentive can be developing some common task through which the participants’ income can later be increased by the sale of the jointly elaborated products [31,38]. 

### 4.2. Strengths and Limitations 

One of the strengths of this review is that recognized guides are employed for its elaboration and analysis of the texts selected, implying an exhaustive and robust review of the available evidence. The authors’ participation in the final consensus selected has allowed contributing data at a high analytical level. 

A limitation of this review is the limited availability of previous randomized experimental trials with a control group and blinding. Most of the studies did not describe in depth the characteristics defining the support groups for breastfeeding, nor did they record the interactions among the participants. Another limitation is missing information on relevant outcomes and details about process evaluation in some studies. More experimental studies with a control group are needed to specifically assess the effectiveness of the measures implemented to foster breastfeeding.

## 5. Conclusions

This systematic review showed that the most successful group strategies to support and maintain breastfeeding during postpartum are those that combined peer support with the leadership or counselling of a health professional or IBCLC. However, the support interventions described in the studies were very heterogeneous in relation to the characteristics of the groups and to the periodicity of the meetings, reason why more studies are needed, randomized and with interventions of similar characteristics, which allow for better data comparison.

## Figures and Tables

**Figure 1 ijerph-18-02550-f001:**
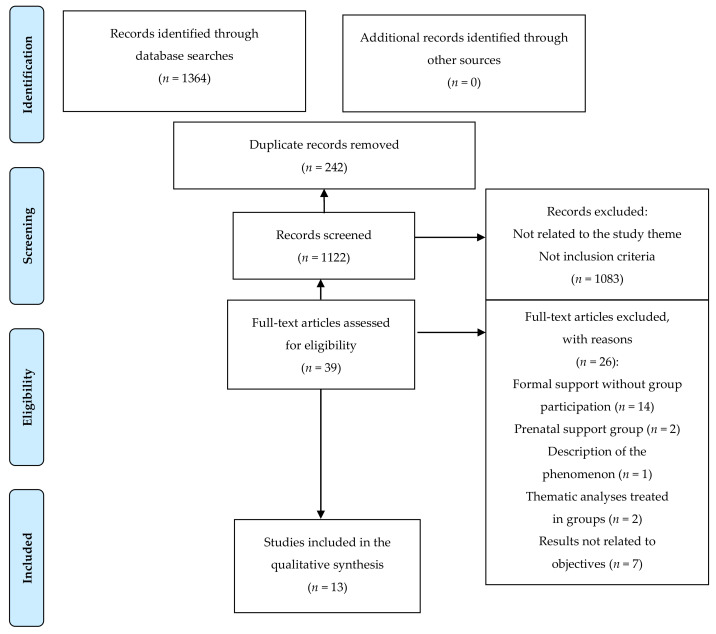
Flow diagram of the selection of articles according to PRISMA.

**Figure 2 ijerph-18-02550-f002:**
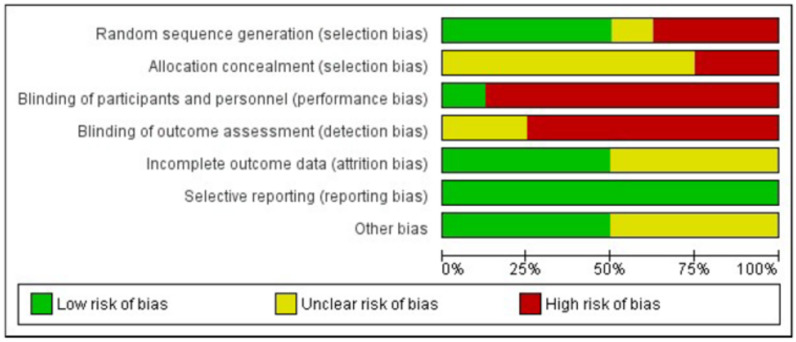
Risk of bias graph-experimental studies.

**Figure 3 ijerph-18-02550-f003:**
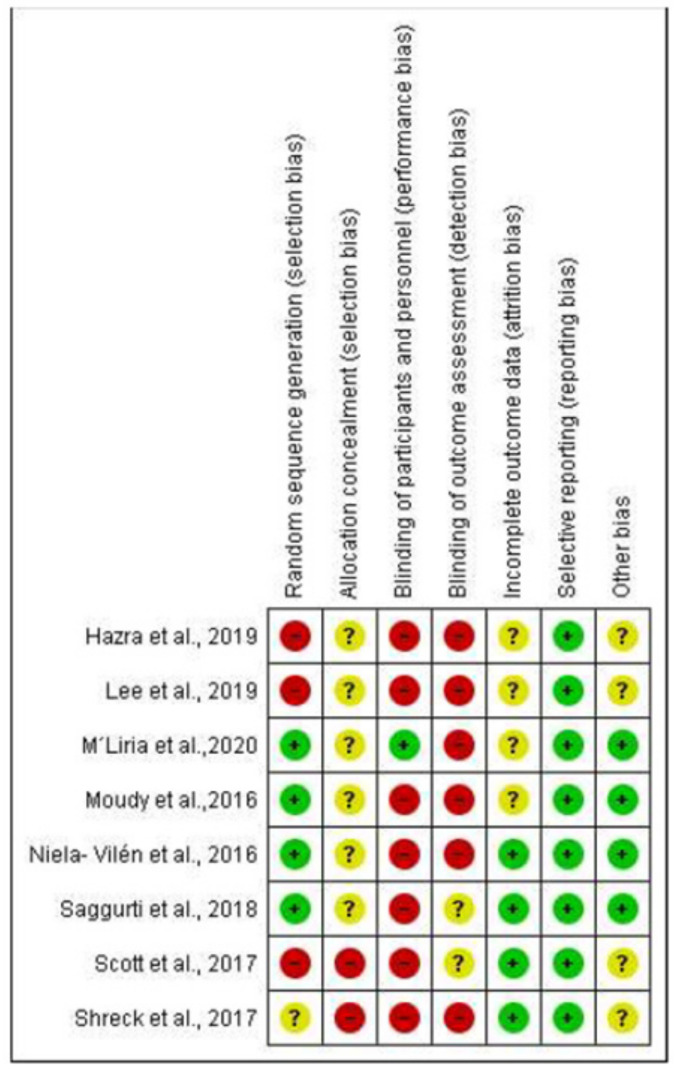
Risk of bias summary-Experimental studies.

**Table 1 ijerph-18-02550-t001:** Search strategy in the databases.

Database	Strategy	Filter	Results
**MEDLINE**	(breastfeeding OR “breast feeding” OR “milk secretion” OR lactation) AND (“support group” OR meeting OR “self-help group” OR “breastfeeding workshop” OR “Group Processes”) AND (impact OR effect OR influence)	2015–2020, humans, language English, Spanish and Portuguese	195
**SCOPUS**		2015–2020, language English, Spanish and Portuguese	193
**WOS**		2015–2020, language English, Spanish and Portuguese	520
**CINHAL**		2015–2020, language English, Spanish and Portuguese	90
**COCHRANE LIBRARY**		2015–2020	363
**LILACS**		2015–2020, language English, Spanish and Portuguese	3

**Table 2 ijerph-18-02550-t002:** Characteristics of the articles included in this review.

Author, Year and Country	Study Design	Objective	Study Characteristics	Intervention	Measuring Instrument	Main Results	Impact on Breastfeeding	Quality Assessment Outcome
Rayfield et al. [36] (2015) United Kingdom	Observational Descriptive Cross-sectional and Comparative study	To explore the association between breastfeeding support and breastfeeding among late preterm (34–36 gestational weeks) and term (≥37 gestational weeks) infants.	Participants = 15,104 singletons born with a gestational age over 34 weeks, of which 14,525 (95.9%) were full term and 579 (4.1%) were late preterm.	Prenatal and postnatal professional advice (midwives or midwife care assistants) + support group Face to face	Structured questionnaire Infant Feeding Survey (IFS) Self-report 6–10 weeks 4–6 months 8–10 months	The mothers who used support from community support groups had significantly higher BF rates than those who did not use this support (*p* = 0.043). The mothers who reported receiving contact details for breastfeeding support groups had a higher likelihood of breastfeeding late preterm (aOR = 3.14, 95% CI = 1.40 to 7.04) and term (aOR = 2.24, 95% CI = 1.86 to 2.68) infants at 10 days and term infants at 6 weeks (aOR = 1.83, 95% CI = 01.51 to 2.22).	Successful breastfeeding	High
Moudi et al. [28] (2016) Iran	Quasi-experimental study	To compare the effect of breastfeeding promotion interventions on exclusive BMF among primiparous women	Participants = 93 primigravidae mothers Age = 18–35 years old Three groups: peer support, education by health care providers, and control (routine prenatal and postpartum care).	The selected mothers in the centre assigned to the peer support were introduced to a volunteer peer with regard to cultural, social, and economical similarities. Face to face and telephone interventions were combined. The selected mothers in the centre assigned to the training health care providers received the first training at 36–38 gestational weeks, and three latter training sessions in 1-, 2- and 3-week intervals after birth by a health care provider. Face to face and telephone interventions were combined.	Structured questionnaire Exclusive breastfeeding rate Initiation–4 weeks–8 weeks	BMF initiation = No significant differences in the three groups (*p* = 0.879) The three groups had no significant difference in terms of duration of exclusive BMF at 4 weeks (*p* = 0.056) but did have a significant difference at 8 weeks (*p* = 0.014) Exclusive BMF at 4 weeks = Peer support group (90.3%), education by health care provider group (83.3%) and control group (65.6%). Significant differences in the three groups (*p* = 0.043). Exclusive BMF at 8 weeks = Peer support group (6.5%), education by health care provider group (6.7%) and control group (28.1%). Significant difference between the three groups (*p* = 0.023).	Successful breastfeeding	1-
Undlien et al. [37] (2016) Kenya	Observational Descriptive Cross-sectional and Comparative study	To determine how the Mother Support Groups (MSGs) affect the nutrition status of children under 2 years of age.	Participants = 20 mothers in each group (41 children) IG: Children whose mothers participated in MSGs CG: Children whose mothers did not participate in MSGs. Both groups received standardized treatment (including supplementary feeding if defined as malnourished) and counselling. Age = Children aged 6 months old or less.	Groups of women either pregnant or with children under 5 years of age learn about the importance of breastfeeding and adequate nutrition by means of health education, demonstrations and discussions. Face to face	Structured questionnaire Every month (during 1 year) First week postpartum (T1) Five-six weeks postpartum (T2)	There was no significant difference between the two groups with respect to breastfeeding practices (*p* = 0.414). Every mother in both groups stated that they had breastfed their child exclusively for the first 6 months.	No significant difference	High
Niela-Vilén et al. [29] (2016) Finland	Randomized controlled trial	To examine whether an Internet-based peer support intervention has any effect on the duration of breastfeeding, breast milk expression or maternal breastfeeding attitude compared with routine care in the mothers of preterm infants.	Participants = 124 mothers (64 in the CG; 60 in the IG) Intervention: Peer-support group in social media (Facebook) Control: Routine breastfeeding support in the NICU	A closed breastfeeding peer-support group in social media (Facebook). Peer support was provided by three voluntary mothers who had previous experience on breastfeeding their own preterm infants. The participating mothers were also peer supporters of each other.	Structured questionnaire Exclusive/ Overall breastfeeding rate Iowa Infant Feeding Attitudes Scale (IFFAS) Breastfeeding Self-Efficacy—Short Form (BSES-SF) 1st week postpartum—Infant’s corrected age of 3/6/12 months	Duration of exclusive BMF; *p* = 0.10 Duration of overall BMF; *p* = 0.60 Duration of expressing milk; *p* = 0.80	No significant difference	1-
Scott et al. [32] (2017) UK	Pre-/Post-quasi-experimental	To evaluate the effectiveness of a Breastfeeding Peer Support Service (BPSS) in increasing breastfeeding initiation and duration in young mothers.	Participants = 5790 women Age < 25 year old	The supporters receive externally accredited BPS training prior to supporting women. Paid peer supporters offer intensive one-to-one support from 30–34 gestational weeks until 6 weeks post-partum, with the highest intensity of support provided during the 2 weeks following birth, offering ongoing and responsive support (face to face or by telephone) according to the women’s individual needs	Any and Exclusive breastfeeding rate Baseline-2 weeks postpartum-6 weeks postpartum	Prevalence at birth increased by 0.55 percentage points per month (95% CI = 0.10–1.00, *p* = 0.018) and, at 2 weeks, by 0.50 percentage points (95% CI = 0.15–0.86, *p* = 0.007). There was no change from an increasing trend in prevalence at 6 weeks.	Successful breastfeeding	1-
Schreck et al. [34] (2017) USA	Pre-/Post- quasi- experimental	To measure the effect of hospital-based prenatal and postnatal breastfeeding interventions on breastfeeding initiation and continuation rates in a low-income population.	Participants: 650 women. Baseline group *n* = 330 Post-intervention group *n* = 320 Age ≥ 18 years old	The prenatal intervention consisted of a breastfeeding-focused prenatal education curriculum offered one-on-one by an IBCLC. The postnatal intervention consisted of a breastfeeding support group.	Structured questionnaire Baseline/Post-intervention	Breastfeeding initiation rates were higher in the post-intervention group (*p* < 0.0001). The mothers in the post-intervention group were significantly more likely to breastfeed (*p* = 0.027, OR = 1.7) compared with those in the baseline group. The breastfeeding continuation rate at or beyond 6 months did not differ between the baseline and post-intervention groups (*p* = 0.5), but was greater among women who also participated in the breastfeeding support group compared with those who participated in the prenatal intervention alone. Over 95% of the participating women reported that the support group was influential in their decision to continue breastfeeding.	Successful breastfeeding	1-
Saggurti et al. [30] (2018) India	Pre-post quasi- experimental	To evaluate a behaviour-changing health intervention with women’s Self-Help Groups (SHGs) aimed at promoting healthy maternal and newborn practices among the more socially and economically marginalized groups.	Participants: 545 SHGs Age ≥ 18 years old Eight sessions	The SHGs included eight weekly cycles of participatory behaviour communication using different thematic modules on maternal, neonatal, child health and promoting collectivization processes facilitated by community health facilitators or sahelis.	Structured questionnaire Baseline-12 months	In the IG, a significant difference is observed in terms of EBF compared to the two measurements (*p* = 0.001); this difference is non-existent in the CG (*p* = 0.22). EBF showed a statistically significant increase over time for SHGs with health integration than without health integration (*p* < 0.05).	Successful breastfeeding	1-
Ruducha et al. [38] (2018) India	Observational, Descriptive Cross-sectional and Comparative study	To expand the understanding of village dynamics in India and how first degree social and advice networks, as well as the cognitive perceptions of 185 women who had recently given birth in areas with and without women’s Self-Help Groups (SHGs), affect immediate breastfeeding.	Participants: 185 women Age = 18–43 years old	Social and advice networks, with important messages related to maternal and newborn health, in microfinance organizations of SHGs. The health workers include Accredited Social Health Activist, voluntary village health worker paid for specific tasks, Auxiliary Nurse Midwife	Structured questionnaire	The women in the SHG blocks had a significantly higher proportion of immediate breastfeeding than those in the non-SHG blocks (66.7% vs. 41.5%, *p* = 0.0010)	Successful breastfeeding	High
Lee et al. [35] (2019) Taiwan	Quasi-experimental	To examine the effectiveness of breastfeeding education and peer support groups organized by International Board Certified Lactation Consultants (IBCLCs)	Participants: 214 postpartum women. CG (*n* = 122): Standard care. IG (*n* = 92): Standard care and support group.	Peer support groups organized by IBCLCs, with weekly face-to-face meetings. Each group consisted of 8–10 mothers.	Breastfeeding Self-Efficacy Scale-Short Form (BSES-SF) Exclusive breastfeeding rate T1: First week postpartum T2: Sixth week postpartum	Exclusive BMF rate at T1 was high and there was no significant proportion difference between groups (*p* = 0.11) Exclusive BMF rate at 6 weeks postpartum = CG: 39%, IG: 61% (*p* = 0.001)	Successful breastfeeding	1-
Hazra et al. [33] (2019) India	Quasi-experimental study	To assess the effects of health behaviour changing interventions through women’s self-help groups (SHGs) on maternal and newborn health behaviours.	Participants: 8865 women. (120 geographic blocks, IG; *n* = 83 blocks, CG). Age = 15 to 49 years old.	The intervention included maternal and child health information dissemination in SHG meetings by trained peer educators, building community norms for behaviour change through a set of community outreach activities including home visits, community meetings, and community events.	Structured questionnaire Round 1: 2015 Round 2: 2017	There are statistically significant differences in both groups at BF initiation (*p* = 0.047), but they were not observed in the duration of EBF (*p* = 0.697). Regarding socioeconomic differences, in the “Least marginalized” group, no differences were observed in both measurements: (*p* = 0.93) and (*p* = 0.19), respectively. However, in the “Most marginalized” group, a difference was observed in breastfeeding initiation (*p* = 0.001), but it was not observed in EBF (*p* = 0. 439).	No significant difference	1-
Robinson et al. [39] (2019) USA	Observational, Descriptive, Cross-sectional and comparative study .	To identify the sources of breastfeeding support for mothers who participate in support groups on Facebook, and to explore a possible mechanism by which the support received on social network sites leads to behavioural outcomes among breastfeeding mothers.	Participants = 277 Afro-American mothers (from 6 breastfeeding online support groups) Age = 19 to 49 years old.	Support groups on Facebook	Structured questionnaire The Network Support for Breastfeeding (NSB) IOWA Infant Breastfeeding Scale (IFFAS) Breastfeeding Self-Efficacy Scale-Short Form (BFSE-SF)	Compared with other support sources, Facebook was valued with a mean score of 2.7 (± 0.38), compared to health care provider support, 2.3 (± 0.74) and spouse support, 2.0 (± 0.78), among others. There were statistically significant differences between the Facebook online support groups and non-Facebook support in terms of the intention of BF duration, perceived self-efficacy and BF attitudes (*p* < 0.05)	Successful breastfeeding	High
M’Liria et al. [31] (2020) Kenya	A cluster randomized controlled trial	To assess the impact of Mother-to-Mother Support Groups on the promotion of exclusive breastfeeding.	Participants = 249 women (Three study groups) IG1: Breastfeeding education and support during seven monthly meetings (MES) IG2: IG1 Intervention + Income generating activities (MESIGA)	Breastfeeding information and support. At each meeting, one topic on breastfeeding was discussed in a session moderated by a trained facilitator. Groups of at most 15 mothers each to facilitate easy sharing of breastfeeding information and mutual support.	Structured questionnaire (WHO/ UNICEF) Baseline- Monthly at 6 months postpartum	The women in the MES and MESIGA groups were twice as likely to exclusively breastfeed at 6 months compared to those in the control group [RR = 2.42; CI = 1.36–4.28; (*p* = 0.004)] and [RR = 1.89; CI = 1.02-3.49; (*p* = 0.033)], respectively. Cumulative EBF was significantly shorter at 0.7 (± 0.15) months in the CG compared to the mean duration of EBF for mothers in the MES (2.8 months) and MESIGA (3.4 months) groups (*p* = 0.001)	Successful breastfeeding	1+
Jenkins et al. [40] (2020) USA	Observational, Descriptive, Cross-sectional and comparative study	To evaluate the “Baby Café” program as a support instrument for breastfeeding.	Participants: 559 women. Melrose Baby Café San Antonio Baby Café	Peer support coupled with professional breastfeeding care (IBCLC) or oversight by an IBCLC when the cafe is staffed with alternative approved breastfeeding counsellors. Regular weekly sessions offering comfortable informal seating, breastfeeding positive messaging and snacks.	Structured questionnaire Baseline–6 months–12 months	Exclusive BMF assistant in Baby Café: 77% (Melrose)/ 52% (San Antonio) at 3 months, 71% (Melrose)/47% (San Antonio) at 6 months. Comparative: 47% at 3 months, 25% at 6 months. Any breastfeeding at 12 months: 67% (Melrose)/56% (San Antonio). Comparative: 36%.	Successful breastfeeding	High

## Data Availability

All data generated or analyzed during this study are included in this published article.

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
