# Peer review of "Impact and Effectiveness of Group Strategies for Supporting Breastfeeding after Birth: A Systematic Review"

_ijerph, 2021, doi:10.3390/ijerph18052550_

Round 1
Reviewer 1 Report
Thank you for the opportunity to review this manuscript. There are many systematic reviews in the field of support for breastfeeding, however, I am not aware of one addressing group strategies specifically. Therefore this review has the potential to add to the body of knowledge on what works to increase breastfeeding rates. Clearly a lot of hard work has gone into this review .
As currently presented the review is not ready for publication mainly due to lack of detail and specification in the methods. The authors have mistakenly used the PRISMA checklist for a systematic review protocol rather than the review checklist. There is no mention of a protocol in the manuscript - there is an expectation that systematic reviews have a protocol that is registered (e.g. with Prospero) or otherwise available for scrutiny. The eligibility criteria, which are of key importance in systematic reviews lack detail. For example:
Participants – how did you define ‘newborns’ – this usually refers to the few days after birth but you have assessed outcome up to 6 months? Infants would be a more accurate term. Did you include only healthy women or those with medical or obstetric complications e.g. those with diabetes, women with HIV, obese women etc. Did you include those with multiple births or singleton births only?
Intervention – you do not mention ‘groups’ in your intervention inclusion criteria - presumably you excluded studies delivered one-to-one. What was your definition of a group – e.g. how many women? Did you include a combination of group and individual support or group only?
Under comparison you have described study design. What comparisons did you include e.g. no additional support, individual support? Did you include comparisons of different types or frequencies of groups?
Outcomes needs to be more specific – did you assess at different timepoints and up until when? Your results include initiation but it is difficult to see how interventions delivered after birth can affect breastfeeding initiation rates?
Study design – please specify more clearly what you mean by observational studies and which designs you included/excluded. Did you include systematic reviews?
What were the exclusion criteria for the review?
The search strategy included a good range of databases - did you search for any grey literature? The search terms address three concepts - it would probably have been sufficient to search on ‘breastfeeding (and all synonyms)’ AND groups (and all synonyms). Adding the third concept of impact/effect/influence may have made the search too specific - i.e. you may have missed some relevant studies.
The text under the heading ‘data extraction’ is in fact a mix of study selection and quality assessment. The data extraction process is missing – please describe what data you extracted and the tool used. What did you do about missing information – especially outcome data e.g. did you contact study authors?
Quality assessment – what was your justification for establishing your own overall scoring system based on the Strobe criteria (you do not report the results of the assessment in the results section) – how did the results of this impact on the review?
The methods should also indicate the planned approach to synthesis - did you plan to conduct meta-analysis if possible and if so what were the criteria for assessing if this was appropriate? In figure 1 you suggest ‘qualitative synthesis’ – narrative synthesis would be a better term so as not to confuse with qualitative research designs which I assume you excluded although you do not state that.
Results
The Prisma flow chart has the same box twice – ‘full text articles assessed for eligibility’ is entered where it should state titles and abstracts. The number of titles and abstracts assessed and exclude should also be indicated in the text. The reasons for exclusion at full text stage are unclear -what is meant by ‘description of the phenomenon and ‘thematic analysis treated in groups’?
Table 2: contains insufficient detail of methods e.g. Rayfield et al – what kind of observational study - cohort/case control/cross-sectional? It would also be helpful to provide information of the intervention – e.g. frequency, duration, dose etc. who provided the intervention and were they trained in breastfeeding support? The table should also contain details of the comparison where relevant.
There are some really helpful details of the characteristics of the included studies but these are quite difficult to read - it might have helped to theme them e.g. who provided the support, frequency, duration and intensity; mode of provision e.g. face to face or online; whether the support groups were combined with other support strategies in multi-component interventions.
Page 14 – what do you mean by ’legal age’ – legal age for what and does this not vary depending on the country? What do you mean by less favoured women – is this referring to income, education, ethnicity are other characteristics that may impact on breastfeeding? Under participants it is helpful to provide the total number of women and infants in the study in your review.
Line 255 – can you specify what you mean by ‘in the immediate postpartum? Do you mean the first day after birth, the first week, on discharge?
A description of comparisons is needed.
The introduction and discussion sections could have been strengthened with wider reference to the abundant literature in the field of breastfeeding support. There are numerous systematic reviews of which the authors should be aware and cite e.g.:
- Haroon S, Das JK, Salam RA, Imdad A, Bhutta ZA. Breastfeeding promotion interventions and breastfeeding practices: a systematic review. BMC Public Health. 2013;13(Suppl 3):S20.
- Shakya P, Kunieda MK, Koyama M, Rai SS, Miyaguchi M, Dhakal S, Sandy S, Sunguya BF, Jimba M. Effectiveness of community-based peer support for mothers to improve their breastfeeding practices: a systematic review and meta-analysis. PLoS One. 2017;12(5):e0177434
- Sinha B, Chowdhury R, Sankar MJ, Martines J, Taneja S, Mazumder S, Rollins, N, Bahl R, Bhandari N. Interventions to improve breastfeeding outcomes: a systematic review and meta-analysis. Acta Paediatr. 2015;104(Supplement 467):114–
- Kim SK, Park S, Oh J, Kim J, Ahn S. Interventions promoting exclusive breastfeeding up to six months after birth: a systematic review and meta-analysis of randomized controlled trials. Int J Nurs Stud. 2018;80:94–
- Olufunlayo TF, Roberts AA, MacArthur C, Thomas N, Odeyemi KA, Price M, Jolly K. Improving exclusive breastfeeding in low and middle-income countries: A systematic review. Matern Child Nutr. 2019;15(3):e12788.
- McFadden A, Siebelt l, Marshall, JL, Gavine A, Girard, L-C, Symon A, MacGillivray S. (2019) Counselling interventions to enable women to initiate and continue breastfeeding: a systematic review and meta-analysis. International Breastfeeding Journal 14: 42. https://doi.org/10.1186/s13006-019-0235-8
These reviews might also provide examples of how to report the methods and findings of reviews, as well enabling the manuscript to demonstrate what new knowledge their review adds. It also seems remiss not to refer to global policy for breastfeeding support - especially the UNICEF/WHO Baby Friendly Hospital Initiative or to the influential Lancet Series on Breastfeeding published in 2016.
The manuscript needs careful editing for English language.
Author Response
Thank you for the opportunity to review this manuscript. There are many systematic reviews in the field of support for breastfeeding, however, I am not aware of one addressing group strategies specifically. Therefore this review has the potential to add to the body of knowledge on what works to increase breastfeeding rates. Clearly a lot of hard work has gone into this review.
Response: Thank you for reviewing the manuscript and for your feedback to improve it.
As currently presented the review is not ready for publication mainly due to lack of detail and specification in the methods. The authors have mistakenly used the PRISMA checklist for a systematic review protocol rather than the review checklist. There is no mention of a protocol in the manuscript - there is an expectation that systematic reviews have a protocol that is registered (e.g. with Prospero) or otherwise available for scrutiny.
Response: It has been changed and sorry for the inconvenience. All the changes in the manuscript have been added coloured in red. Please see, Supplementary Table S1: PRISMA 2009 Checklist: Recommended items in a systematic review. Reference number 23 was modified in the bibliography. According to your suggestion of registration with PROSPERO, we would like to say that we tried to do it, but systematic reviews not related to COVID-19 are not admitted so far as you can read:
"To enable PROSPERO to focus on COVID-19 registrations during the 2020 pandemic, this registration record was automatically rejected because it did not meet the acceptance criteria."
The eligibility criteria, which are of key importance in systematic reviews lack detail. For example:
Participants – how did you define ‘newborns’ – this usually refers to the few days after birth but you have assessed outcome up to 6 months? Infants would be a more accurate term. Did you include only healthy women or those with medical or obstetric complications e.g. those with diabetes, women with HIV, obese women etc. Did you include those with multiple births or singleton births only?
Intervention – you do not mention ‘groups’ in your intervention inclusion criteria - presumably you excluded studies delivered one-to-one. What was your definition of a group – e.g. how many women? Did you include a combination of group and individual support or group only?
Under comparison you have described study design. What comparisons did you include e.g. no additional support, individual support? Did you include comparisons of different types or frequencies of groups?
Outcomes needs to be more specific – did you assess at different timepoints and up until when? Your results include initiation but it is difficult to see how interventions delivered after birth can affect breastfeeding initiation rates?
Study design – please specify more clearly what you mean by observational studies and which designs you included/excluded. Did you include systematic reviews?
Response: Thank you for your feedback for improving this important part in a review. The items in the PICO question have been clarified. All the changes in the manuscript have been included in red, please see the lines 85-102.
What were the exclusion criteria for the review?
Response: The inclusion and exclusion criteria have been clarified. They are described in the lines 115-126.
The search strategy included a good range of databases - did you search for any grey literature? The search terms address three concepts - it would probably have been sufficient to search on ‘breastfeeding (and all synonyms)’ AND groups (and all synonyms). Adding the third concept of impact/effect/influence may have made the search too specific - i.e. you may have missed some relevant studies.
Response: We agree that the last term used in the search strategy is very specific, however, it was useful to define the large volume of results generated by that search only with the combination of the other terms. The authors searched also in grey literature including Google and Google Scholar, but no relevant results were obtained. Please see the line 83-84;130-131.
The text under the heading ‘data extraction’ is in fact a mix of study selection and quality assessment. The data extraction process is missing – please describe what data you extracted and the tool used. What did you do about missing information – especially outcome data e.g. did you contact study authors?
Response: The present review followed the guidelines given in the Preferred Reporting Items for Systematic Reviews and Meta-Analyses (PRISMA) Statement, using a three-step procedure to identify relevant studies (lines 79-81, Figure 1). We have used Mendeley® reference manager, to import all references and delete duplicate citations. Please see the lines 128-130.The authors have only assessed the information described in the studies and studies´ authors were not contacted.
Quality assessment – what was your justification for establishing your own overall scoring system based on the Strobe criteria (you do not report the results of the assessment in the results section) – how did the results of this impact on the review?
Response: For the quality assessment of the articles performed with an observational design, the statement of the Strengthening the Reporting of Observational studies in Epidemiology (STROBE) initiative was used as a reference in the absence of a tool specified for experimental studies. The authors agreed to the valuation percentages based on other assessments described in the literature and reported the results in table 2. The quality assessment of these studies was taken into account as an exclusion criterion when a low assessment was found.
The methods should also indicate the planned approach to synthesis - did you plan to conduct meta-analysis if possible and if so what were the criteria for assessing if this was appropriate? In figure 1 you suggest ‘qualitative synthesis’ – narrative synthesis would be a better term so as not to confuse with qualitative research designs which I assume you excluded although you do not state that.
Response: Yes, the authors valued the possibility of a meta-analysis, but due to the high heterogeneity of characteristics of the participating populations, as well as to the interventions carried out, it was not possible to perform a meta-analysis. A narrative analysis of the results was conducted. This is specified in the text, please see the lines 140-144.
We also thought that the narrative term would be more appropriate, however, we have followed structure and nomenclature recommended by PRISMA 2009 Flow diagram, avaible in: http://prisma-statement.org/documents/PRISMA%202009%20flow%20diagram.pdf
Results
The Prisma flow chart has the same box twice – ‘full text articles assessed for eligibility’ is entered where it should state titles and abstracts. The number of titles and abstracts assessed and exclude should also be indicated in the text. The reasons for exclusion at full text stage are unclear -what is meant by ‘description of the phenomenon and ‘thematic analysis treated in groups’?
Response: PRISMA flow format have been changed now, sorry for the inconvenience. Please, see Figure 1: Flow diagram of the selection of articles according to PRISMA. We have also included it in the text the number of titles and abstracts exclude. All the changes in the manuscript have been modified in red. Please see the lines 179-191.
In relation to “description of the phenomenon” and “thematic analysis treated in groups”, is described in 2.2. Inclusion/Exclusion criteria (Please, see the lines 116-126).
It refers to articles found after the search that referred to the description of groups, from a psychosocial perspective, understanding them as mutual self-helps groups. For example this following article: Baño, I.;Carrillo, C.; Thambidurai, U.; Martínez, M. E. El concepto del Baby Café como red internacional de apoyo a la lactancia materna. Cultura de los Cuidados, 2015,19: 43. Avaible online :< http://dx.doi.org/10.14198/cuid.2015.43.03>
Table 2: contains insufficient detail of methods e.g. Rayfield et al – what kind of observational study - cohort/case control/cross-sectional? It would also be helpful to provide information of the intervention – e.g. frequency, duration, dose etc. who provided the intervention and were they trained in breastfeeding support? The table should also contain details of the comparison where relevant.
There are some really helpful details of the characteristics of the included studies but these are quite difficult to read - it might have helped to theme them e.g. who provided the support, frequency, duration and intensity; mode of provision e.g. face to face or online; whether the support groups were combined with other support strategies in multi-component interventions.
Changes have been made to Table 2, including details of observational studies as well as interventions made.
Page 14 – what do you mean by ’legal age’ – legal age for what and does this not vary depending on the country? What do you mean by less favoured women – is this referring to income, education, ethnicity are other characteristics that may impact on breastfeeding? Under participants it is helpful to provide the total number of women and infants in the study in your review.
Response: Legal age refers to the recognized age of majority in each country, so we specify that this concept refers to the specific sample of each study.
Less favoured population has been defined in references 30-32, 34 and 38, as population with less socioeconomic resources, social and economically marginalized groups.
The total number of women has been added. Please see the line 278. It is not possible to add the total number of infants, because some studies do not define whether the sample of women included single or multiple births.
Line 255 – can you specify what you mean by ‘in the immediate postpartum? Do you mean the first day after birth, the first week, on discharge?
A description of comparisons is needed.
Response: The authors of the studies refer to the immediate postpartum measurement in the first 24 hours after birth. It has been modified in the text, please see the lines 304-305.
The introduction and discussion sections could have been strengthened with wider reference to the abundant literature in the field of breastfeeding support. There are numerous systematic reviews of which the authors should be aware and cite e.g.:
- Haroon S, Das JK, Salam RA, Imdad A, Bhutta ZA. Breastfeeding promotion interventions and breastfeeding practices: a systematic review. BMC Public Health. 2013;13(Suppl 3):S20.
- Shakya P, Kunieda MK, Koyama M, Rai SS, Miyaguchi M, Dhakal S, Sandy S, Sunguya BF, Jimba M. Effectiveness of community-based peer support for mothers to improve their breastfeeding practices: a systematic review and meta-analysis. PLoS One. 2017;12(5):e0177434
- Sinha B, Chowdhury R, Sankar MJ, Martines J, Taneja S, Mazumder S, Rollins, N, Bahl R, Bhandari N. Interventions to improve breastfeeding outcomes: a systematic review and meta-analysis. Acta Paediatr. 2015;104(Supplement 467):114–
- Kim SK, Park S, Oh J, Kim J, Ahn S. Interventions promoting exclusive breastfeeding up to six months after birth: a systematic review and meta-analysis of randomized controlled trials. Int J Nurs Stud. 2018;80:94–
- Olufunlayo TF, Roberts AA, MacArthur C, Thomas N, Odeyemi KA, Price M, Jolly K. Improving exclusive breastfeeding in low and middle-income countries: A systematic review. Matern Child Nutr. 2019;15(3):e12788.
- McFadden A, Siebelt l, Marshall, JL, Gavine A, Girard, L-C, Symon A, MacGillivray S. (2019) Counselling interventions to enable women to initiate and continue breastfeeding: a systematic review and meta-analysis. International Breastfeeding Journal 14: 42. https://doi.org/10.1186/s13006-019-0235-8
These reviews might also provide examples of how to report the methods and findings of reviews, as well enabling the manuscript to demonstrate what new knowledge their review adds. It also seems remiss not to refer to global policy for breastfeeding support - especially the UNICEF/WHO Baby Friendly Hospital Initiative or to the influential Lancet Series on Breastfeeding published in 2016.
Response: We think so too, some of the references recommended were used in the manuscript before, please see:
Reference 1: World Health Organization.Exclusive Breastfeeding.Available online: https://www.who.int/nutrition/topics/exclusive_breastfeeding/es/ (accesed on 25 September 2020).
Reference 2: World Health Organization, UNICEF. Global strategy for infants and young child feeding.World Health Organization: Ginebra, 2003. Available online: https://www.who.int/maternal_child_adolescent/documents/9243562215/es/ (accesed on 25 September 2020).
Reference 13: World Health Organization, UNICEF. Protecting, promoting, and supporting breastfeeding in facilities providing maternity and newborn services: the revised Baby-friendly Hospital Initiative.WHO: Switzerland, 2018. Available online: https://www.who.int/nutrition/publications/infantfeeding/bfhi-implementation/en/ (accesed on 28 September 2020).
Reference 43: Kim, S.; Park, S.; Oh, J.; Kim, J.; Ahn, S. Interventions promoting exclusive breastfeeding up to six months after birth: a systematic review and meta-analysis of randomized controlled trials. Int. J. Nurs. Stud. 2018, 50, 94–105.
However, following your suggestion, we've also incorporated some of the recommended references:
Reference 42: Olufunlayo,T.F.; Roberts, A.A.; MacArthur C, Thomas, N;, Odeyemi, K.A.; Price, M.; et al. Improving exclusive breastfeeding in low and middle-income countries: A systematic review. Matern Child Nutr. 2019,15 :e12788
Reference 48: Shakya, P.; Kunieda, M.K.; Koyama, M.; Rai, S.S.; Miyaguchi, M.; Dhakal, S.; et al. Effectiveness of community-based peer support for mothers to improve their breastfeeding practices: a systematic review and meta-analysis. PLoS One. 2017,12:e0177434
Reference 49: Sinha, B.; Chowdhury, R.; Sankar, M.J; Martines, J., Taneja, S.; Mazumder, S.; et al. Interventions to improve breastfeeding outcomes: a systematic review and meta-analysis. Acta Paediatr,2015,104:114-134. doi: 10.1111/apa.13127. PMID: 26183031
Citations in the text when inserting new bibliography have undergone changes, are described in red in the text.
The manuscript needs careful editing for English language.
Response: Attached you may find the certification of editing and English translation by a professional translator.
Reviewer 2 Report
REVIEWER´S COMMENTS
GENERAL
This systematic review carried out by the authors aims to analyse the most effective breastfeeding group support practices performed during postpartum, as well as the characteristics associated to their success in maintaining breastfeeding for a longer period.
However, throughout the document and for a better understanding of the reader, not all revised interventions in the works selected in the final sample are accurately described and classified. Also, little said of the characteristics associated with long-term breastfeeding maintenance.
SPECIFICS
Line 97. Inclusion/Exclusion criteria.
This section is not clarified. The inclusion and exclusion criteria must be well defined independently. It would be convenient to clarify the limitation that was made by type of study in database searches. I suggest to describe in detail which are the support groups, and to clarify if other influencing factors have been considered in the initiation and maintenance of breastfeeding.
It would help a lot to understand the work to write the research question that the researchers asked themselves at the beginning of the study.
The description of the components of the research question (PICO) is not well defined. It is not always relevant to include “Comparison” in the PICO strategy. It is not appropriate to include it if it is not compared with another intervention. In any case, "Comparison" are not the types of studies included in the work.
Line 110
What does the author mean by “phenomenon from the social perspective or the Support groups”?
Clarify in detail the different inclusion / exclusion criteria
Line 178. Figure 1.
The number and title should be written below the figure. The number and title of a table (in italics) is written above it. The table describing "Records excluded: Not related to the study theme (n = 817)" should be extended from the table above containing n=849 articles There are Exclusion criteria that are described, that are not mentioned in the text. Please clarify them in methodology.
Researchers are advised to present larger text boxes in this figure so that the sentences are not cut.
Line 203 and line 205. Figure 2 and 3
The number and title of a figure is written below it. Keep the journal’s recommendations for font size and font. Titles of tables and figures. The legend below the figure should not be in bold and it is recommended to incorporate this phrase in the text of the manuscript not in the image of the Figure 3 is recommended a better presentation of the text to facilitate its reading and visibility and also incorporate the phrase that describes the content of the figure in the text of the manuscript, not in the image of the figure.
Line 207.
Perhaps it would be better to just write "characteristics of the sample"
Line 211 and 212
You should not start a sentence with a number.
Line 213
Clarify the phrase.
Line 223
Clarify what the authors mean by "impact of breastfeeding". It refers to the results of interventions delivering significant results in breastfeeding, but to what extent? Longer? Higher percentage? Both or either?
Line 228. Table 2
Correct the text format of the "Measuring Instrument" column. There are words that are cut off
In Table 2, page 9, Article 37. MSGs is quoted but the term has not been described above. The first time it is done is in line 314. However, the author rewrites the description and the acronym in line 467
Table 2, page 13, the term Mother-to-Mother Support Groups is shortened with the acronym MTMSGs but would not be necessary as this term is not requoted in the text.
Line. 229 Participants
What does the author mean by "the impact of the community and online support groups"? You mean Breastfeeding? Clarify, please.
Line 232, 270
In this manuscript the main theme is the strategies carried out for the promotion of breastfeeding. This paragraph refers to different types of breastfeeding support services carried out in the selected studies. Would this, broadly speaking, be the same as breastfeeding peer Support (BPSS)? They are not described in this work previously in the section of material and methods, but are cited throughout the manuscript and somewhat more in-depth information is reported in results. However, it does not clarify what each type of intervention or service consists of, and different terminology is used to refer to them. In addition, there are other terms such as MSGs that are not quoted in the results section (text from line 280). I think it is important to make a brief description of each type, to clarify what each type consists of at the beginning of the work. This could help to better understand the manuscript. Subsequent partial clarifications of a particular type of Support (line 329) should have been done with all support types previously.
Line 279, 423
No description of acronyms is required again. This term (IBCLC) was previously specified in Table 2.
Line 339
The term describing SHGs has already been described above. It should only be written with its acronyms.
Line 522
Line 369 cites a study that combines both Support interventions
Discussion
There is information in this section that could have been included in the results section such as the different interventions carried out with online strategies (from line 535)
It should be mentioned in results and in discussion what are the "the characteristics associated to their successs in maintaining BF for a longer period of time", as it is one of the objectives of this work.
References
The title of references 1, 2, 3, 6, 7 and 21 should be translated into English. In references 1, 2 and 6 the authorship as well.
Access data missing in Reference 27, 30, 37
Author Response
GENERAL
This systematic review carried out by the authors aims to analyse the most effective breastfeeding group support practices performed during postpartum, as well as the characteristics associated to their success in maintaining breastfeeding for a longer period.
However, throughout the document and for a better understanding of the reader, not all revised interventions in the works selected in the final sample are accurately described and classified. Also, little said of the characteristics associated with long-term breastfeeding maintenance.
Response: Thank you for your appreciation. Wide changes were included in the manuscript and this request was amended.
SPECIFICS
Line 97. Inclusion/Exclusion criteria.
This section is not clarified. The inclusion and exclusion criteria must be well defined independently. It would be convenient to clarify the limitation that was made by type of study in database searches. I suggest to describe in detail which are the support groups, and to clarify if other influencing factors have been considered in the initiation and maintenance of breastfeeding.
Response: Inclusion and exclusion criteria have been reformulated. Please, see lines 116-126.
Support groups, in the selection of studies does not show a common structure, this is the reason why it is impossible to establish a clear comparison or meta-analysis. Please see the lines, 66-73 ;140-144; 312-423 and 627-632.
It would help a lot to understand the work to write the research question that the researchers asked themselves at the beginning of the study.
Response: Regarding this request, authors consider that the aim of the systematic review give response to this issue.
The description of the components of the research question (PICO) is not well defined. It is not always relevant to include “Comparison” in the PICO strategy. It is not appropriate to include it if it is not compared with another intervention. In any case, "Comparison" are not the types of studies included in the work.
Response: Thank you for your review of the manuscript and for your feedback to improve it. These changes have been implemented in the text in red, also in accordance with the requested by reviewer 1. Please see the lines 85-102 and lines 116-126.
Line 110
What does the author mean by “phenomenon from the social perspective or the Support groups”?
Response: It refers to articles found after the search that explained the description of aid groups, from a psychosocial perspective, understanding them as mutual self-helps groups. For example, this article: Baño, I.;Carrillo, C.; Thambidurai, U.; Martínez, M. E. El concepto del Baby Café como red internacional de apoyo a la lactancia materna. Cultura de los Cuidados, 2015,19: 43. Avaible online :< http://dx.doi.org/10.14198/cuid.2015.43.03>
Clarify in detail the different inclusion / exclusion criteria
Response: These changes have been implemented in the text in red. Please see the lines, 116-126.
Line 178. Figure 1.
The number and title should be written below the figure. The number and title of a table (in italics) is written above it. The table describing "Records excluded: Not related to the study theme (n = 817)" should be extended from the table above containing n=849 articles There are Exclusion criteria that are described, that are not mentioned in the text. Please clarify them in methodology.
Researchers are advised to present larger text boxes in this figure so that the sentences are not cut.
Response: Errors in the PRISMA flow format have been changed, sorry for the inconvenience. Please see Figure 1: Flow diagram of the selection of articles according to PRISMA. We have also included it in the text the number of titles and abstracts excluded. All the changes in the manuscript have been modified in red. Please see the lines 179-191.
Line 203 and line 205. Figure 2 and 3
The number and title of a figure is written below it. Keep the journal’s recommendations for font size and font. Titles of tables and figures. The legend below the figure should not be in bold and it is recommended to incorporate this phrase in the text of the manuscript not in the image of the Figure 3 is recommended a better presentation of the text to facilitate its reading and visibility and also incorporate the phrase that describes the content of the figure in the text of the manuscript, not in the image of the figure.
Response: It has been modified, please, see Figure 2 and 3, and the lines 247-250..
Line 207.
Perhaps it would be better to just write "characteristics of the sample" .
Response: It has been changed. Please, see the line 255.
Line 211 and 212
You should not start a sentence with a number.
Response: It has been changed. Please, see the line 259.
Line 213
Clarify the phrase.
Response: It has been changed. Please, see the line 259-262.
Line 223
Clarify what the authors mean by "impact of breastfeeding". It refers to the results of interventions delivering significant results in breastfeeding, but to what extent? Longer? Higher percentage? Both or either?
Response: It refers to the results of interventions delivering significant results in breastfeeding, such as higher percentage or longer throughout time, than comparison group. Please, see the lines 268-270.
Line 228. Table 2
Correct the text format of the "Measuring Instrument" column. There are words that are cut off
In Table 2, page 9, Article 37. MSGs is quoted but the term has not been described above. The first time it is done is in line 314. However, the author rewrites the description and the acronym in line 467.
Response:It has been changed.
Table 2, page 13, the term Mother-to-Mother Support Groups is shortened with the acronym MTMSGs but would not be necessary as this term is not requoted in the text.
Response: It has been changed.
Line. 229 Participants
What does the author mean by "the impact of the community and online support groups"? You mean Breastfeeding? Clarify, please.
Response: It refers to community groups. It has been clarified in the text. Please see the line 281.
Line 232, 270
In this manuscript the main theme is the strategies carried out for the promotion of breastfeeding. This paragraph refers to different types of breastfeeding support services carried out in the selected studies. Would this, broadly speaking, be the same as breastfeeding peer Support (BPSS)? They are not described in this work previously in the section of material and methods, but are cited throughout the manuscript and somewhat more in-depth information is reported in results. However, it does not clarify what each type of intervention or service consists of, and different terminology is used to refer to them. In addition, there are other terms such as MSGs that are not quoted in the results section (text from line 280). I think it is important to make a brief description of each type, to clarify what each type consists of at the beginning of the work. This could help to better understand the manuscript. Subsequent partial clarifications of a particular type of Support (line 329) should have been done with all support types previously.
Response:
Breastfeeding Peer Support (BPS) it is refers to support mothers to mothers. The aim is to improve breastfeeding rates within a community. Breastfeeding peer support aims to suppoprt mothers who want to breastfeed with others who had breastfed successfully. Peer support is recommended by the World Health Organization and is a part of UK NICE guidance on maternal and child nutrition and on routine postnatal care [1,2]. However, Breastfeeding Peer Support Service (BPSS) is the specific support service offered by the Health System, and may be different from one city to another city [3,4].
This review has included any type of support for group breastfeeding, including peers only, groups with or without leadership from a healthcare professional, with a IBCLC leadership, a peer volunter… Without restrictions on group variability or regularity of the meetings or leadership. It is defined in the eligibility criteria PICO format, please see the lines 93-99.
- The Baby Friendly Iniative. Available online: https://www.unicef.org.uk/babyfriendly/breastfeeding-peer-support-what-works/#:~:text=Breastfeeding%20peer%20support%20aims%20to,and%20on%20routine%20postnatal%20care. (Acessed on 9 febryary 2021).
- Breastfeeding peer support service.Available online: https://www.nct.org.uk/about-us/commissioned-services/breastfeeding-peer-support-training (Acessed on 9 february 2021).
- Notthinghramshire Healthcare. NHS Foundation Trust. Breastfeeding and infant feeding support. Available online: https://www.nottinghamshirehealthcare.nhs.uk/infant-feeding (Acessed on 9 february 2021)
- Ingram, J. A mixed methods evaluation of peer support in Bristol, UK: mothers’, midwives’ and peer supporters’ views and the effects on breastfeeding. BMC Pregnancy Childbirth13, 192 (2013). https://doi.org/10.1186/1471-2393-13-192
The characteristics of the interventions are heterogeneous, all this information is described in 3.6 Characteristics of the interventions, please, see the lines 312-423. Also included in Table 2.
Line 279, 423
No description of acronyms is required again. This term (IBCLC) was previously specified in Table 2.
Response:It has been changed.
Line 339
The term describing SHGs has already been described above. It should only be written with its acronyms.
Response:It has been changed.
Line 522
Response:It has been changed.
Line 369 cites a study that combines both Support interventions
Response: Niela- Vilén et al. [29] combined peer support with formal support online. Please see the lines 411-417.
Discussion
There is information in this section that could have been included in the results section such as the different interventions carried out with online strategies (from line 535)
It should be mentioned in results and in discussion what are the "the characteristics associated to their successs in maintaining BF for a longer period of time", as it is one of the objectives of this work.
Response: This information is described in lines 424-553, 572-580 and 606-610..
References
The title of references 1, 2, 3, 6, 7 and 21 should be translated into English. In references 1, 2 and 6 the authorship as well.
Response: References 1 and 2 have been changed. References 3,6,7 and 21 are originally in Spanish, this is the reason why we did not translate them into English.
Access data missing in Reference 27, 30, 37.
Response: It has been clarified in red in the references.
Reviewer 3 Report
A novel area of research, methodology elaborately described and limitations well stated and conclusion provides area for more research in Exclusive breastfeeding in the first six months.
There was no mention of the effect of marketing of Breast milk substitutes and the working mothers with limited enabling environment for supportive breastfeeding. These are worth exploring in future researches.
Author Response
A novel area of research, methodology elaborately described and limitations well stated and conclusion provides area for more research in Exclusive breastfeeding in the first six months.
There was no mention of the effect of marketing of Breast milk substitutes and the working mothers with limited enabling environment for supportive breastfeeding. These are worth exploring in future researches.
Thank you for your review of the manuscript and for your feedback to improve it. We will consider your feedback for future research.
Reviewer 4 Report
Ijerph-1087111
Thank you for the opportunity to review this manuscript.
This is a systematic review that aims to analyse the effectiveness of interventions to support breastfeeding after birth.
Although the manuscript largely conforms to the methodology of systematic reviews, there are some modifications that must be carried out, preventing its publication in its current state, which I will now detail.
1.- Title: It should be adapted to the effectiveness of the support strategies.
2.- Abstract: must be separated into the following headings: background, methods, results and conclusion.
3.- Methodology: The search strategy section should be improved, since there is repeated information in the manuscript and in Table 1. I recommend creating a new table that includes the PICO question, the keywords and the different databases, data consulted with the results obtained. Feel free to use an example published by the same journal doi: 10.3390 / ijerph17207405
4.- Authors must explain why they have not searched the Cochrane Database.
5.- In my opinion, in the PICO question, there is no comparison, because the type of studies is one of the filters that should be used regarding the type of study. So, I recommend removing it from the text and putting “Not applicable”.
6.- The authors must explain why there are 520 articles in WOS, and 195 in Medline or Scopus. Do not include this explanation in the manuscript. In addition, 6 databases are detailed in the abstract and in Figure 1, only 5 are shown.
7.- In a systematic review, studies that are not randomized or quasi-experimental clinical trials should not be included, because if we want to measure the effect of the intervention, we will need two branches to obtain conclusive results. Therefore, I recommend only including in the SR the 8 articles with the most scientific evidence (2 RCT and 6 QE), to which an analysis of biases has been applied.
8.- Due to the previous comment, adapt the sections of results, discussion and conclusions, including the modification of Figure 1, ans supplementary tables.
Author Response
Thank you for the opportunity to review this manuscript. This is a systematic review that aims to analyse the effectiveness of interventions to support breastfeeding after birth.
Although the manuscript largely conforms to the methodology of systematic reviews, there are some modifications that must be carried out, preventing its publication in its current state, which I will now detail.
Response: Thank you for your review of the manuscript and for your feedback to improve it.
1.- Title: It should be adapted to the effectiveness of the support strategies.
Response: The comment has been considered. Please, see the tittle.
2.- Abstract: must be separated into the following headings: background, methods, results and conclusion.
Response: Considerations for publication have been followed in the instructions for authors :“ The abstract should be a total of about 200 words maximum. The abstract should be a single paragraph and should follow the style of structured abstracts, but without headings “. Available from: https://www.mdpi.com/journal/ijerph/instructions
3.- Methodology: The search strategy section should be improved, since there is repeated information in the manuscript and in Table 1. I recommend creating a new table that includes the PICO question, the keywords and the different databases, data consulted with the results obtained. Feel free to use an example published by the same journal doi: 10.3390 / ijerph17207405
Response: Thanks for this observation. The items in the PICO question have been clarified in the text, please see the lines 85-103. Table 1 shows additional information about the individual results obtained in each database, according to item number 8 PRISMA 2009 Checklist: Recommended items in a systematic review.
4.- Authors must explain why they have not searched the Cochrane Database.
Response: Considering exclusively the inclusion of original studies in the review, the search was not performed in this database in a consensual decision by the research team. The authors considered that the five selected databases, including MEDLINE and WOS, could cover all information required.
5.- In my opinion, in the PICO question, there is no comparison, because the type of studies is one of the filters that should be used regarding the type of study. So, I recommend removing it from the text and putting “Not applicable”.
Response: Thank you for your feedback for improving this important part in a review. The items in the PICO question have been clarified. All the changes in the manuscript have been modified in red, please see the lines 85-102.
6.- The authors must explain why there are 520 articles in WOS, and 195 in Medline or Scopus. Do not include this explanation in the manuscript. In addition, 6 databases are detailed in the abstract and in Figure 1, only 5 are shown.
Response: Attending your request we performed the search again with same filters, obtaining similar results.
The error in the databases has been changed, we apologize for the inconvenience.
7.- In a systematic review, studies that are not randomized or quasi-experimental clinical trials should not be included, because if we want to measure the effect of the intervention, we will need two branches to obtain conclusive results. Therefore, I recommend only including in the SR the 8 articles with the most scientific evidence (2 RCT and 6 QE), to which an analysis of biases has been applied.
8.- Due to the previous comment, adapt the sections of results, discussion and conclusions, including the modification of Figure 1, ans supplementary tables.
Response: Thank you for your feedback to improve the manuscript. We agree with you that quasi-experimental studies and clinical trials are the best methodological designs for measuring the effectiveness of interventions, however, given the nature of the topic, many authors defined the impossibility of carrying out these methodological designs by opting for observational study. Version 5.1.0 of the Cochrane handbook emphasizes that, in many situations, it is not practical or possible to blind participants or study staff in the intervention group (Higgins JPT, Green S (editors). Cochrane Handbook for Systematic Reviews of Interventions Version 5.1.0 [updated March 2011]. The Cochrane Collaboration, 2011. Available from www.cochrane-handbook.org.). In this way, we considered that these results also provided important information to the review.
In addition, reviewer 1 has requested more details on these studies. In other systematic reviews published not only in this journal but others, these kinds of studies were included, for example:
López-González, Á.; García-Quintanilla, M.; Guerrero-Agenjo, C.M.; Tendero, J.L.; Guisado-Requena, I.M.; Rabanales-Sotos, J. Eficacy of Cryotherapy in the Prevention of Oral Mucosistis in Adult Patients with Chemotherapy. Int. J. Environ. Res. Public Health 2021, 18, 994. https://doi.org/10.3390/ijerph18030994
Pérez-Ordás, R.; Nuviala, A.; Grao-Cruces, A.; Fernández-Martínez, A. Implementing Service-Learning Programs in Physical Education; Teacher Education as Teaching and Learning Models for All the Agents Involved: A Systematic Review. Int. J. Environ. Res. Public Health 2021, 18, 669. https://doi.org/10.3390/ijerph18020669
Olufunlayo TF, Roberts AA, MacArthur C, Thomas N, Odeyemi KA, Price M, Jolly K. Improving exclusive breastfeeding in low and middle-income countries: A systematic review. Matern Child Nutr. 2019,15(3):e12788.
Round 2
Reviewer 2 Report
I think the authors have gone to great lengths to implement the changes and the recommendations.
The manuscript has improved a lot.
Author Response
I think the authors have gone to great lengths to implement the changes and the recommendations.
The manuscript has improved a lot.
Response: Dear reviewer, thank you for this positive feedback.
Round 2
Thanks for your input. There are different comments that I suggest making:
Material and methods:
1.- They must clarify through which platform each of the databases, OVID SP, EBSCO, etc., has been consulted.
2.- They must identify the acronyms of the authors who have carried out the peer review, and which author resolved the disagreements.
3.- Each of the consulted databases needs a different structure when carrying out the search strategy. Thus, a common structure reported in Table 1 is not valid. The previous comment made, which has already been made in different publications of this journal, has not been taken into account, and provides greater clarity and detail for the reader.
4.- The authors have not resolved comment 6 of the first round, in which they requested an explanation of the discrepancy in terms of results obtained in the WOS database with respect to the other sources.
5.- I need more information about the reason for the exclusion of the 817 articles, in which the authors only indicate that they were not related to the topic. It is difficult to be rejected in the screening phase because the consistency of the search would not be adequate, or it would lack a well-structured formula. It is more common to find in this phase as reasons for exclusion that the type of study is not adequate, for example.
6.- Although it is true that the argument that the authors make in comment 8 of the first round is partially correct, I strongly insist that the inclusion of observational studies should not have been carried out because there is sufficient bibliography with greater evidence to analysis of the impact of support groups. In fact, they have included different systematic reviews carried out in their bibliography (42, 43, 48 and 49). From here to establish a causal relationship, there is no doubt that the randomized clinical trials may indicate a cause-effect and the observational study may not do so in any case.
7.- They must justify by means of a bibliographic reference on which the quality categories awarded to the STROBE scale are based (high quality / medium-low / low).
8.- Based on my proposals, the sections of results, discussion and conclusions should be adapted.
Author Response
Thanks for your input. There are different comments that I suggest making:
Response: Thank you for reviewing the manuscript and for your feedback to improve it.
Material and methods:
1.- They must clarify through which platform each of the databases, OVID SP, EBSCO, etc., has been consulted.
Response: Thank you for this comment. Now, this is included Please, see line 84.
2.- They must identify the acronyms of the authors who have carried out the peer review, and which author resolved the disagreements.
Response: It has been modified in red. Please, see the lines (129;135) . Due to the high degree of concordance (Kappa statistics = 0.81), a third author was not needed to resolve disagreements.
3.- Each of the consulted databases needs a different structure when carrying out the search strategy. Thus, a common structure reported in Table 1 is not valid. The previous comment made, which has already been made in different publications of this journal, has not been taken into account, and provides greater clarity and detail for the reader.
Response: The search strategy designed was verified and tested in the different databases, reporting a significant number of articles in each of them. Therefore, the same search strategy was used in all databases, performing a systematized, clearly defined and explicit search.
4.- The authors have not resolved comment 6 of the first round, in which they requested an explanation of the discrepancy in terms of results obtained in the WOS database with respect to the other sources.
Response: Although PUBMED is an outstanding database in the international field of health sciences,that comprises more than m 32 million citations for biomedical literature from MEDLINE, WOS is also a powerful multi-discipline database, both scientific, sociological or humanistic, which can lead to more results given the research the topic in this database. Currently, main collection WOS contains nearly 18.245 high-quality international journal publications and MEDLINE 5.243 journals. You can replicate search strategy with the filters to observe the difference in results.
5.- I need more information about the reason for the exclusion of the 817 articles, in which the authors only indicate that they were not related to the topic. It is difficult to be rejected in the screening phase because the consistency of the search would not be adequate, or it would lack a well-structured formula. It is more common to find in this phase as reasons for exclusion that the type of study is not adequate, for example.
Response: In the screening phase, the 1.083 articles excluded refer to the total results obtained in each database, which after reading the title and abstract, was not related to the study objectives. PRISMA sets in this step the total number of records or citations deleted, prior to full-text reading. Please, see the lines 185-187.
6.- Although it is true that the argument that the authors make in comment 8 of the first round is partially correct, I strongly insist that the inclusion of observational studies should not have been carried out because there is sufficient bibliography with greater evidence to analysis of the impact of support groups. In fact, they have included different systematic reviews carried out in their bibliography (42, 43, 48 and 49). From here to establish a causal relationship, there is no doubt that the randomized clinical trials may indicate a cause-effect and the observational study may not do so in any case.
Response: The research team agrees that the best methodology for analyzing the impact of interventions is experimental studies, specifically randomized clinical trials, this consideration is exposed in strengths and limitations (please see the lines 634-635). However, taking into account the literature reported in the previous round, also that much of the biomedical research is observational should not therefore be negligible, and the methodology according to the STROBE tool of the observational articles included, the decision to include them was maintained.
7.- They must justify by means of a bibliographic reference on which the quality categories awarded to the STROBE scale are based (high quality / medium-low / low).
Response: To the quality assessment of the observational studies, the research team used the STROBE scale as a reference for critical appraisal and interpretation.This is only a checklist of 22 items, which relate to the title, abstract, introduction, methods, results and discussion sections of articles. The STROBE provides guidance to authors about how to improve the reporting of observational studies and facilitates critical appraisal and interpretation of studies by reviewers, journal editors and readers, but it does not establish a scoring system to evaluate studies already carried out, so the research topic agrees this score based on the ítems.
8.- Based on my proposals, the sections of results, discussion and conclusions should be adapted.
Response: recommendations that were amended, were included in the proper section.